# The Applicability of Nanostructured Materials in Regenerating Soft and Bone Tissue in the Oral Cavity—A Review

**DOI:** 10.3390/biomimetics9060348

**Published:** 2024-06-08

**Authors:** Giorgiana Corina Muresan, Sanda Boca, Ondine Lucaciu, Mihaela Hedesiu

**Affiliations:** 1Department of Oral Health, Iuliu Hatieganu University of Medicine and Pharmacy, 400012 Cluj-Napoca, Romania; giorgianamuresan@yahoo.com; 2Interdisciplinary Research Institute in Bio-Nano-Sciences, Babes-Bolyai University, 400271 Cluj-Napoca, Romania; sanda.boca@ubbcluj.ro; 3Department of Oral Radiology, Iuliu Hatieganu University of Medicine and Pharmacy, 400006 Cluj-Napoca, Romania; mhedesiu@gmail.com

**Keywords:** nanostructures, oral cavity, tissue engineering, bone engineering

## Abstract

Background and Objectives: Two of the most exciting new technologies are biotechnology and nanotechnology. The science of nanostructures, or nanotechnology, is concerned with the development, testing, and use of structures and molecules with nanoscale dimensions ranging from 1 to 100 nm. The development of materials and tools with high specificity that interact directly at the subcellular level is what makes nanotechnology valuable in the medical sciences. At the cellular or tissue level, this might be converted into focused clinical applications with the greatest possible therapeutic benefits and the fewest possible side effects. The purpose of the present study was to review the literature and explore the applicability of the nanostructured materials in the process of the regeneration of the soft and hard tissues of the oral cavity. Materials and Methods: An electronic search of articles was conducted in several databases, such as PubMed, Embase, and Web of Science, to conduct this study, and the 183 articles that were discovered were chosen and examined, and only 22 articles met the inclusion criteria in this review. Results: The findings of this study demonstrate that using nanoparticles can improve the mechanical properties, biocompatibility, and osteoinductivity of biomaterials. Conclusions: Most recently, breakthroughs in tissue engineering and nanotechnology have led to significant advancements in the design and production of bone graft substitutes and hold tremendous promise for the treatment of bone abnormalities. The creation of intelligent nanostructured materials is essential for various applications and therapies, as it allows for the precise and long-term delivery of medication, which yields better results.

## 1. Introduction

The oral cavity is susceptible to a variety of biological, physical, chemical, and mechanical stimuli [1]. Hard and soft tissues in the oral cavity form an ideal environment for microbial development and biofilm formation, making it prone to a variety of oral pathological conditions, such as dental caries and pulpitis, periodontal disease, and inflammatory and tumoral lesions of the oral mucosa [1].

Nanotechnology is undoubtedly one of the key technologies of the new millennium [2]. Nanotechnologies contribute to almost every field of science, aiming to obtain systems and materials with unique features and, in recent years, being applied to improve human health with promising results [3]. In dentistry, nanotechnology has vast applications in diagnostics, dental materials, preventive dentistry, dental aesthetics, endodontics, regenerative dentistry, periodontology, implantology, etc. [4]. One definition of nanotechnology is a technique that works with small materials or structures [4]. Nanostructures or nanomaterials are defined as smaller than 100 nm in one size, dimensions that give them unique physicochemical properties. The types of nanostructures more common in dentistry are nanoparticles, nanotubes, nanofibers, nanocomposites, hydrogel, antimicrobial materials, mineralization, and coatings [4].

The biomedical applications of nanostructured materials have received considerable attention, and various nanomaterials have been recently developed for biomedical applications [1]. One such application is tissue engineering, which is generated by the knowledge of engineering and sciences and aims to obtain tissues and other biological superstructures that are similar to those present in the body and help to maintain, improve, or recover the function of various affected organs [5]. A successful tissue engineering process depends on manipulating gene expression and cellular interactions, and hence, selecting a scaffold material that closely mimics native tissue structure is of paramount importance for restoring relative function [5].

For the oral cavity, tissue engineering aims at the regeneration of functional tissues, with the help of nanostructured scaffolds that provide signaling molecules and cells [2]. Specifically, it allows for the controlled growth of bone and periodontal tissues with the use of scaffolds, cells, and signaling molecules [6]. In this regard, the application of nanomaterials and stem cells in tissue regeneration is a new emerging field with great potential for maxillofacial bone defects [6]. Nanostructured scaffolds provide structural support closer to natural bone, while stem cells allow for the regeneration of bone tissue in places in which a certain volume of bone is crucial to achieving successful implantation [6].

Various biomaterials are used in oral tissue engineering to obtain three-dimensional scaffolds that promote cell adhesion, proliferation, and differentiation and could improve oral bone regeneration [7] Biomaterials used in tissue engineering can be of natural or synthetic origin and are able to come into immediate contact with living tissue without developing any adverse immune reaction [7]. The placement of these new biomaterials in the affected area triggers a series of events, inducing regenerative cellular responses and resulting in the replacement of the missing tissue [7].

In order to reproduce as faithfully as possible the nanostructure of the natural extracellular matrix (ECM), scaffolds made of nanofibers, nanotubes, nanoparticles, and hydrogel have emerged as promising materials [8]. Since tissues/organs have nanometer dimensions and cells directly interact with nanostructured ECMs, the biomimetic characteristics and excellent physicochemical properties of nanomaterials play an essential role in stimulating cell growth, as well as in guided tissue regeneration [8].

The aim of this research was to provide a more detailed picture of recent advances in nanotechnology used in regenerative therapy in the oral cavity, as well as new techniques for obtaining nanostructured materials. 

## 2. Materials and Methods

Focused Review Question: 

Are nanostructured materials an option in regenerating bone and soft tissue from the oral cavity?

The PICO question was structured as follows:

Population (P): Bone and soft tissue defects that require regeneration;

Intervention (I): Regenerative therapy using nanostructured materials;

Comparison (C): Nanostructured materials and classical therapies;

Outcome (O): Regeneration of bone and soft tissue. 

Inclusion and exclusion criteria are as follows: 

In order to carry out this narrative review of the specialized literature, an examiner conducted an electronic search of the most relevant articles published in English between January 2018 and May 2023 in the following databases: PubMed, Embase, and Web of Science.

The inclusion criteria included articles describing the use of nanostructures in soft and hard tissue regeneration, as well as articles presenting new techniques for obtaining these nanomaterials, in vitro studies on animal models, or randomized controlled studies that studied various nanostructures with applicability in the regeneration of soft and hard tissues of the oral cavity.

The exclusion criteria were reviews and meta-analyses, articles that were not on the topic, articles for which the full text could not be read, and non-English articles.

Search and Screening strategy:

The search strategy included the following MESH terms: “nanostructure”, “bone regeneration”, and “oral cavity”. In addition, the following filters were applied: articles published in the last 5 years for all three databases used and, in the Embase database, articles only from Embase, to remove duplicates.

Data Extraction and analysis:

All results were uploaded into Rayyan software, and duplicates were removed. Following this, the 2 authors (CGM, SB) independently screened abstracts and titles using the Rayyan software. Any articles were included or excluded based on agreement via discussion between the two evaluators. A full-text screening was then conducted by both examiners with the aim of checking the eligibility of the articles included in this study. Any disagreements were resolved by re-evaluating the articles or with the help of a third person (OL).

The Preferred Reporting Items for Systematic Reviews and Meta-Analyses (PRISMA) flow diagram (Figure 1) was used to illustrate the search method and format the manuscript [9].

In order to assess the quality of the studies included in this review, we used the quality assessment tool for in vitro studies (QUIN instrument), and the results were included in Table 1. The QUIN tool includes 12 criteria with the following scoring and grading options: adequately specified (score = 2), inadequately specified (score = 1), not specified (score = 0), or not applicable (X); this allows clinicians to evaluate the quality of in vitro studies. The scores thus obtained were used to grade the in vitro study as high-, medium-, or low-risk (>70% = low risk of bias, 50% to 70% = medium risk of bias, and <50% = high risk of bias) by using the following formula: final score = (total score × 100)/(2 × number of criteria applicable) [10]. 

## 3. Results

The preliminary search returned 183 articles, of which 149 were found in PubMed, 32 were found in Embase, and 2 were found in Web of Science. The application of the filters led to the selection of 60 articles, which were further scanned by title and abstract, leading to the exclusion of another 15 articles. As a result of this sorting, 45 articles remained in the selection group, which were examined by the full text and led to the exclusion of another 23 articles that were not on our topic. Finally, 22 articles remained eligible to be included in this literature search (Figure 1).

Of the 22 articles included in this review, 1 article is based on an animal model, and 21 are in vitro studies. Of the in vitro studies included in this review, nine articles also included animal studies in their research. Of these 22 studies included in the research, 8 articles assessed the effects of bone regeneration of nanostructured materials, 7 articles evaluated their effects in terms of the treatment of periodontal diseases, more precisely of soft tissues, 2 articles assessed the anti-inflammatory and antibacterial effects, and 3 highlighted the effects that materials have on bone and tissue regeneration together.

All the studies included in this research aimed to develop new techniques/materials based on nanostructures that can be used in various pathologies of the oral cavity (Table 2).

Applications of nanostructures in bone regeneration.

Membranes are essential in regenerative therapy. They prevent the penetration of connective and epithelial components at the level of the bone defect, thus playing the role of barrier, but they can also stimulate the regeneration process through substances integrated into their structure. Thus, Moonesi Rad R. et al. [14] created a two-layer membrane containing bioactive glass modified with 7% boron using an electrospun process to prepare the CA/GEL/7B-BG fiber layer and a solvent-cast membrane layer made of CA. This bi-layered membrane, which was obtained using electrospinning, was first made by the Moonesi Rad R. et al. team in 2019, and the research results show that this membrane has an asymmetry, both structurally and in its composition; this asymmetry gives it favorable properties, such as osteoinductivity and bioactivity, thus having potential in bone and soft tissue regeneration [14]. 

Together with his team in 2020, Ghavimi M.A. [23] developed an asymmetric membrane loaded with curcumin and aspirin, with applications in guided bone regeneration. One side of the membrane contains aspirin nanoparticles included in collagen nanofibers, and the other side of the membrane contains collage-curcumin nanofibers. Through in vitro and in vivo studies, the authors demonstrated the beneficial effects that nanostructured membranes have, in addition to the anti-inflammatory effects that are already known; aspirin shows osteoinductive effects, and curcumin shows osteoconductive effects, thus enhancing the effects of aspirin [23].

Another nanostructured scaffold is also used in regenerative therapies. In their research, Hokmabad V.R. et al. [12], identified a new technique that combines the process of electrospinning and freeze-drying in making scaffolds. Through this new technique, the authors developed a new EC-g-PCL/alginate scaffold that exhibits controlled porosity and a surface that mimics the structure of the extracellular matrix, to which they added HA to improve its qualities; this scaffold shows potential for bone regeneration [12].

A new technique for obtaining nanostructured scaffolds is also proposed by Covarrubias C. et al. [19]. In their study, the authors obtained a method of in situ polymerization in one step of the scaffold composed of PU and nBG. This nanostructure, according to the authors’ results, exhibits enhanced bioactive properties to stimulate bone tissue regeneration [19].

Xia Y. et al. [20] obtained a nanostructured scaffold in their research by mixing CPC and IONP powders and confirmed the qualities of this scaffold in terms of bone regeneration through an in vitro study. In their research, Ma L. et al. [26] tested, both in vitro and in vivo, BBR/PCL/COL scaffolds with different concentrations of berberine, the results of which indicated potential use in bone regeneration therapy [20,26].

Boda S.K. et al. [15] obtained mineralized nanofiber fragments from PCG electrospun nanofiber membranes that were frozen and cryocut into segments of 20 μm thickness. These segments were loaded with E7-BMP-2 peptides. In their study, the authors demonstrated that the use of cryocut mineralized nanofiber fragments coupled with calcium-bone morphogenetic protein 2 (BMP-2) showed favorable results in terms of bone regeneration, and they allowed for sustained peptide release for 4 weeks [15]. 

Ou Q. et al. [18] described, in their study, the realization of a zein/gelatin/nHAp nanofibrous membrane. This membrane shows good biocompatibility and osteoinductive behavior and, by including nHAp in the nanofibrous structure of the membrane, could promote hPDLSC adhesion, proliferation, and osteogenic differentiation. The membrane properties were confirmed by both in vitro and in vivo studies [18]

Shi Z. [31] and his team designed and manufactured a hydrogel loaded with nanoparticles of carboxymethyl chitosan/sodium alginate and nanohydroxyapatite called FHCS. The material has thermosensitivity, degradability, and injectability characteristics, finding applicability in bone regeneration therapies [31].

Continuous research and development of nanotechnology has allowed us to obtain membranes with dual capacities, having effects on both hard and soft tissue. Boda S.K. et al. [21], in 2020, created a chitosan membrane that is selectively modified, on one side or partially, with a mucoadhesive polysaccharide layer that gives the membrane the property of adhering to the hard and soft tissues of the oral cavity. The double adhesive function of the membrane makes it useful in the regeneration of soft and hard tissues, and if it is also associated with drug administration, it can also provide antimicrobial protection [21].

In order to accomplish osteointegration and biological sealing, which preserves the implants’ long-term stability, dental implants must be integrated with both gingival and bone tissues [24]. Because of its high mechanical strength and elastic modulus, which are similar to those of human cortical bone and allow it to avoid the stress shielding effects associated with titanium (Ti) based implants, PEEK has recently attracted increased interest as dental implants. PEEK is unable to connect with host bone and promote bone regeneration [27]. To address this shortcoming, Pang Z. et al. [24] and Xie D. et al. [27] performed bioactive coating on PEEK, thus improving surface performance. Pang Z. et al. have used nanostructured coating with TP, obtaining a strong response from RBMS cells and HGE cells when exposed to PKTP [24]. Xie D. et al. [27] used, in their study, FSL (80 mW and 160 mW) to modify the PEEK surface by creating submicro-nano structures to cause cellular exciting, which would induce bone and gingival regeneration for the long stability of dental implants [27].

b.Applications of nanostructures in soft tissue regeneration

In addition to bone regeneration, membranes containing nanostructures also find application in periodontal regeneration. Lam L.R.W. et al. [25] present, in their research, a new method of manufacturing nanostructured membranes and membranes containing EMD encapsulated in core-shell nanofiber. The membrane, according to the authors, is a multifunctional barrier capable of releasing active substances, with sustained release over 22 days and a role in the regeneration of the lost periodontium [25].

Nanoparticles and nanospheres are also used in regenerative therapy, especially periodontal tissues. Three articles included in this article study nanoparticles, namely Ni C. et al. [13], and demonstrate that AuNPs with a diameter of 45 nm enhanced cementogenesis and osteogenesis processes while reducing osteoclastogenesis activity. A periodontal fenestration defect model in rats provided additional confirmation of this conclusion [13]. Zhang Y. et al. [28] analyzed the effects of AuNPs regarding the osteogenic differentiation of stem cell sheets of the periodontal ligament [28]. Xue Y. et al. [17] investigated, both in vitro and in vivo, the optimal combination of three components, chitosan, PLGA, and AgNPs, for periodontal tissue regeneration. Research results indicate that 3:7 nPLGA/nCS and 50 μg/mL nAg ratios represent the optimal ratio to achieve positive effects with low cytotoxicity in periodontal tissue regeneration [17].

Also, membranes, by their composition or structure, can have anti-inflammatory and antimicrobial effects. Chen P. et al., in 2018 [11], and Liu X. et al., in 2020 [22], created nanostructured membranes with anti-inflammatory and antimicrobial effects. In order to confer the anti-inflammatory and antimicrobial effects of the membrane, Chen P. et al. loaded the collagen membrane with AgNP in an optimal concentration and low cytotoxicity [11]. Liu X. et al. made a membrane with a time-programmed release of therapeutic agents [22].

## 4. Discussion

Nanotechnology has produced and is still producing tools for creating biomaterials and pharmaceutical formulations that can significantly increase the efficacy and surgical approaches of pharmaceuticals by taking advantage of and modifying supramolecular materials at the nanometric scale [33].

Therefore, nanostructured materials find use in a variety of therapeutic settings, including the oro-dental field. They can be used as carriers or deliverers of pharmaceuticals with targeted or prolonged release, as well as providing structural support that is more akin to natural tissue and allows for the insertion of various substances. They can also be applied in tissue engineering, bone and soft tissue regeneration, and the treatment of periodontal diseases [33].

Delivery of medicines:

According to this review, the most common indication of nanostructures is as a drug delivery system for drugs with anti-inflammatory and antibacterial effects.

The drawback of systemic medicine delivery is that a small amount may accumulate at the target location, whereas local medicine administration may experience a shorter duration of action as a result of oral cavitary liquids rinsing the medication [34]. Nanostructured materials come to solve these shortcomings, having the advantage of protecting the active substance from local clearance and ensuring the sustained release over a period of time of therapeutic agents [34]. Active substances can be found in nanostructured materials, either encapsulated in nanospheres or nanotubes or present in membrane structures; the drug release rate is dependent on several factors, including the types of materials from which they are made (natural/synthetic), the type of nanostructure, and even the host structure/tissue [34,35].

Nanostructured membranes represent the most common form of nanostructured materials, as they might contain nanofibers, which are further able to support nanoparticles loading the active substance, as described by Chen P. et al. and Chernova U. et al. [11,36]; moreover, the therapeutic agent can be embedded into their structure, according to research by Liu X. et al. [22].

Nanofiber-containing membrane structures can be obtained from ceramic materials, metal compounds, and synthetic polymers through electrospinning, phase separation, self-assembly, or laser spinning. The main advantages of nanofibers are their surface area and porosity, the modification of which can regulate the release of the drug [37].

Nanoparticles in the membrane structure can be organic and consist mainly of lipids, proteins, and inorganic materials, which are mainly crystallized inorganic materials and amorphous solids [37].

2.Soft tissue and bone regeneration

In this area, articles are more numerous and diversified. In our bibliographic search, we have encountered the applicability of various forms of nanostructures, such as nanostructured membranes, nanoparticles, nanotubes, nanofibers, hydrogels, etc. For effective bone and soft tissue regeneration, drug delivery systems and cell differentiation carriers are needed [38]. Nanostructured materials used in regenerative therapies should stimulate tissue regeneration and gradually disintegrate, giving way to newly formed tissue and not triggering immunoreactions [38].

Biodegradable polymers have gained ground due to their rapid and localized absorption, being used to obtain nanofibers from membrane structures and scaffolds. Hydroxyapatite, tricalcium phosphate, and bioactive glass, representing the inorganic component of these nanostructures, have a role in improving osteoblastic adhesion, the differentiation of mesenchymal stem cells and progenitor cells, and angiogenesis [39].

Nanostructured membranes, another type of nanostructured material that is used in bone/soft tissue regeneration, have the following roles: to act as a barrier, to prevent soft tissues from entering the space necessary for the formation of new bone, to eliminate pathogenic bacterial flora, to destabilize the biofilm present, and to favor cell adhesion [40]. The membranes can be made of natural biomaterials, such as chitosan, and synthetic materials—polymers—such as polylactic acid (PLA), poly(glycolic acid) (PGA), poly(lactic-co-glycolic) acid (PLGA), poly(ethylene glycol) (PEG), and polycaprolactone (PCL) [41], and can be loaded with various nanoparticles, such as HA, Ag., etc., aiming to improve biocompatibility and osteoinductivity and facilitating the tissue adhesion, antimicrobial properties, and mechanical properties of the membrane [18,21,40].

The nanofibers presented in this review presented various morphologies, including tubular and sleeve-core, which are obtained using various techniques, the most common being coaxial and conventional electrospinning. These nanofibers are obtained from various substances and arranged in membranes, having positive effects either only on bone regeneration [23] or fulfilling a dual role—bone and soft tissue regeneration [14,15,18,22].

Nanoparticles, as well as nanostructured materials that find applicability in bone and soft tissue regeneration, are included in composite materials and are mainly inorganic metallic nanoparticles (Au, Ag) and metal oxide (IONP) [35]. Regarding AuNPs enhance the osteogenesis of PDLSC sheets [26], AgNPs have antimicrobial action, mainly due to the gradual release of silver ions [16,17,42], AgNPs encapsulated in collagen favored MSC proliferation, osteogenic differentiation, and calcium mineralization [35], and IONPs show excellent biocompatibility and promising effects on osteogenic differentiation and stem cell biomineralization [20,43].

Hydrogel is another category of nanostructured material that attracts interest in bone regeneration. Gelatin, alginate, and hyaluronic acid are the most commonly used natural hydrogels, while polyethyllinglycol is a synthetic hydrogel. An ideal hydrogel for bone regeneration should be easy to make, injectable, biocompatible, and degradable and release the appropriate active growth factors within 2–4 weeks, according to existing research in the literature [31,42].

Polymeric nanofibrous scaffolds are promising candidates for soft tissue regeneration due to their design flexibility and biocompatibility, allowing products with physical properties similar to extracellular matrix [44]. The most common method of obtaining these polymeric nanofibers is electrospinning (conventional, coaxial, etc.) [44]. To prevent possible damage to these polymer scaffolds, many researchers, including Hokmabad V.R. et al. [12], incorporated calcium phosphate and bioactive glass into the polymer matrix, obtaining composite scaffolds that show improved biological properties and osteoinducing effects [39]. Many types of inorganic materials can be integrated into nanofibers present in the structure of these scaffolds, which improve their mechanical properties, and the combination of polymers and inorganic substances, such as hydroxyapatite, is commonly used to increase the biological properties of nanofibrous scaffolds [44,45].

Nanoparticles represent another category of nanostructured materials, which find application in soft tissue regeneration due to their distinct physicochemical properties conferred by the high surface–volume ratio [39]. Ni C. et al. [13] demonstrated, in their research, that gold nanoparticles represent real success in the early regulation of the inflammatory response of periodontal tissues, thus playing an important role in the regeneration of periodontal tissues. AuNPs can interact with MSCs and facilitate osteogenesis, but they can also serve as cell probes for tracking MSCs in vivo [39].

The main limitation of this review was the lack of clinical trials on patients, such studies being found in studies older than five years (one of the filters applied to database search being articles from the last five years). Another limitation of this review was the heterogeneity of the articles, as follows: different methods, techniques used in obtaining and loading nanostructured materials, application techniques, etc.

As shown above, to assess the risk of bias, the QUIN tool was used to assess the items as follows: >70% = low risk of bias, 50% to 70% = medium risk of bias, and <50% = high risk of bias. Six articles included in this review have scores of 73% = low risk of bias, and the vast majority of the articles included in this review have scores ranging from 60% to 68% = medium risk of bias.

## 5. Conclusions

As we observed herein, nanostructured materials find application in regenerative therapy of the oral cavity, in which they either play the role of stimulators of bone and soft tissue regeneration or are used as a means of controlled drug delivery. Nanostructured materials are the future of bone and soft tissue regeneration therapy in the oral cavity. Further research is needed to develop a predictable technology for obtaining these nanostructures.

## Figures and Tables

**Figure 1 biomimetics-09-00348-f001:**
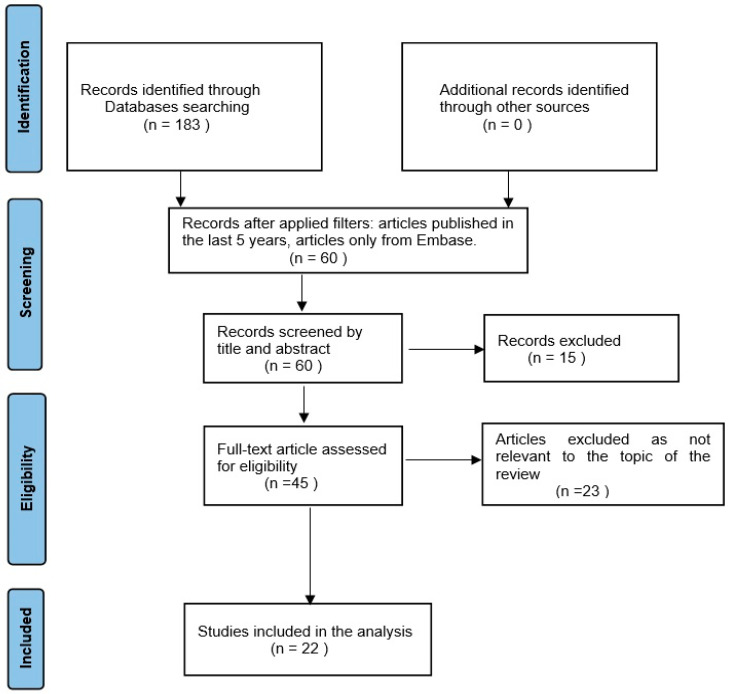
PRISMA flow diagram: diagram representing the inclusion of studies.

**Table 1 biomimetics-09-00348-t001:** QUIN tool.

Author [Reference]	Clearly Stated Aims/ Objectives	Detailed Explanation of Sample Size Calculation	Detailed Explanation of Sampling Technique	Details of Comparison Group	Detailed Explanation of Methodology	Operator Details	Randomization	Method of Measurement of Outcome	Outcome Assessor Details	Blinding	Statistical Analysis	Presentation of Results	Final Score
Chen P. [11]	2	1	1	1	2	0	0	2	0	X	2	2	60%
Hokmabad VR. [12]	2	1	1	1	2	0	0	2	0	X	2	2	60%
Ni C. [13]	2	1	1	2	2	1	2	2	0	X	2	2	68%
Moonesi Rad R. [14]	2	1	1	1	2	1	1	2	1	X	2	2	73%
Boda SK. [15]	2	0	2	2	2	0	1	2	0	X	2	2	68%
Martínez-Sanmiguel JJ. [16]	2	1	1	1	2	1	1	2	1	X	2	2	73%
Xue Y. [17]	2	1	1	1	2	0	1	2	0	X	2	2	63%
Ou Q. [18]	2	1	2	1	2	0	1	2	0	X	2	2	68%
Covarrubias C. [19]	2	1	1	1	2	0	1	2	0	X	2	2	63%
Xia Y. [20]	2	1	1	1	2	0	1	2	0	X	2	2	63%
Boda SK. [21]	2	1	1	1	2	1	1	2	1	X	2	2	73%
Liu X. [22]	2	1	1	1	2	1	1	2	1	X	2	2	73%
Ghavimi MA. [23]	2	2	1	1	2	1	1	2	1	X	1	2	73%
Pang Z. [24]	2	1	1	1	2	0	0	2	0	X	2	2	60%
Lam LRW. [25]	2	1	1	1	2	1	0	2	1	X	2	2	68%
Ma L. [26]	2	1	1	1	2	1	1	2	1	X	2	2	73%
Xie D. [27]	2	1	1	1	2	0	0	2	1	X	2	2	63%
Zhang Y. [28]	2	1	1	1	2	0	0	2	0	X	2	2	60%
Su T. [29]	2	1	1	1	2	0	0	2	0	X	2	2	60%
Kim SY. [30]	2	1	1	1	2	0	0	2	0	X	2	2	60%
Shi Z. [31]	2	1	1	1	2	1	0	2	1	X	2	2	68%

The Quin tool evaluates the quality and risk of bias of in vitro studies. A total of 12 criteria are used to assess quality, which can be assigned one of the following options: adequately specified (score = 2), inadequately specified (score = 1), not specified (score = 0), or not applicable (X). The risk of bias is calculated using the formula mentioned above, obtaining a percentage that will fit the article into one of the following categories: >70% = low risk of bias, 50% to 70% = medium risk of bias, and <50% = high risk of bias.

**Table 2 biomimetics-09-00348-t002:** Characteristics of the articles included in the review.

Author, Year, [Reference]	Study Type and Design	Study Objectives	Nanostructure Type	New Techniques for Obtaining Nanostructures	Applications in Dentistry	Outcomes
Chen P. et al., 2018 [11]	In vitro	- a new collagen membrane coated with silver nanoparticle (AgNP)	- AgNP - coated collagen membrane	- preparation of silver-coated collagen membrane - sonication coating - sputtering coating	- antibacterial and anti-inflammatory capacity	- With only minimal cytotoxicity, the collagen coated with AgNP demonstrated the ability to strengthen the membrane’s antibacterial and anti-inflammatory properties.
Hokmabad V.R. et al., 2019 [12]	In vitro	- obtain new scaffolds of ethyl cellulose-grafted-poly (ε-caprolactone)(EC-g-PCL) obtained by a new manufacturing method	- the new EC-g-PCL/alginate scaffolds with different contents of nanohydroxyapatite	- synthesis, preparation, and characterization of the scaffolds	- tissue and bone engineering	- the hDPSCs exhibited strong adhesion, proliferation, and differentiation on EC-g-PCL/alginate scaffolds combined with nanohydroxyapatite.
Ni C. et al., 2019 [13]	In vitro and in vivo on an animal model	- the potential therapeutic application of gold nanoparticles (AuNPs)—the optimum size	- AuNPs	- not applicable	- tissue and bone engineering	- The 45 nm AuNPs might control the early inflammatory response of periodontal tissues, in addition to directly modulating hPDLCs.
Moonesi Rad R. et al., 2019 [14]	In vitro	– a new asymmetric bilayered membrane	- boron-modified bioactive glass (7B-BG)	- preparation of the bilayered membranes by the electrospinning method and characterization	- bone regeneration	- for GBR applications, a functionally graded bilayered membrane was effectively created.
Boda S.K. et al., 2019 [15]	In vitro and in vivo on a animal model	- the potential of mineralized nanofiber segments coupled with calcium-binding bone morphogenetic protein 2 (E7-BMP-2 peptides)	- mineralized nanofiber segments coupled with E7-BMP-2 peptides	- electrospinning -mineralization - characterization	- bone regeneration	- A potential substitute for bone tissue regeneration is provided by the nanofiber fragments.
Martínez-Sanmiguel J.J. et al., 2019 [16]	In vitro	- fabrication of Hydroxyapatite/silver (HA/Ag) nanocomposites with better antimicrobial efficiency and anti-inflammatory properties	- HA/Ag nanocomposites	- synthesis and characterization of HA/Ag nanocomposites	- antimicrobial and anti-inflammatory capacity	- All of the HA/Ag doses examined (250–7 lg/mL) exhibited antibacterial action against E. Coli and antifungal efficacy against Candida albicans.
Xue Y. et al., 2019 [17]	In vitro and in vivo on a animal model	- investigation of the optimal composite ratio of these three materials for periodontal tissue regeneration	- the mixture of poly(lactic-co-glycolic acid)/chitosan/Ag nanoparticles	- production of nanoparticles of chitosan (CS), poly(lactic-co-glycolic acid) (PLGA), and silver	- tissue regeneration	- The ideal ratio and cell mineralization were facilitated by the nPLGA/nCS/nAg combination, which exhibited no cytotoxicity, as follows: 3:7 nPLGA/nCS and 50 µg/mL nAg ratios.
Ou Q. et al., 2019 [18]	In vitro and in vivo on an animal model	- a new zein/gelatin/nanohydroxyapatite (zein/gelatin/nHAp) nanofibrous membranes	zein/gelatin/nanohydroxyapatite nanofibrous membranes (zein/gelatin/nHAp)	- the electrospun zein/gelatin/nHAp nanofibers	-tissue and bone engineering	- The zein/gelatin/nHAp nanofibers help hPDLSCs adhere, proliferate, and differentiate into osteoblasts.
Covarrubias C. et al., 2019 [19]	In vitro and in vivo on a animal model	- a new method of preparation of bionanocomposite scaffolds	- bionanocomposite scaffolds based on aliphatic polyurethane (PU) and bioactive glass nanoparticles (nBG)	- the one-step in situ polymerization method	- bone regeneration	- PU nanocomposite scaffolds that support bone regeneration can be produced using a one-step in situ preparation technique when nBG particles are present.
Xia Y. et al., 2019 [20]	In vitro	- the effects of the new composite on bone matrix formation and osteogenic differentiation	- iron oxide nanoparticle-calcium phosphate cement (CPC + IONP)	- fabrication and testing of CPC + IONP scaffold	- bone regeneration	- Incorporating IONP into CPC scaffold remarkably enhanced the spreading, osteogenic differentiation, and bone mineral synthesis of stem cells.
Boda S.K. et al., 2020 [21]	In vitro	- the presentation of dual soft mucosal and hard bone/enamel tissue adhesive nanofiber membranes	- dual oral tissue adhesive nanofiber membranes	- fabrication and characterization of the oral dual tissue adhesive	- tissue and bone engineering	- fabrication of the chitosan-based nanofiber membranes with dual adhesion to soft and hard tissue surfaces and pH-controlled delivery of antimicrobial agents, antibiotics, and peptides
Liu X. et al., 2020 [22]	In vitro and in vivo on an animal model	- getting a time-programmed multi-drug releasing system	- core-shell nanofiber membrane	- preparation and characterization of the core-shell nanofiber membrane	- delivery of medicines	- dual-drug core-shell nanofiber membrane with the capacity to control the release order of different drugs
Ghavimi M.A. et al., 2020 [23]	In vitro and in vivo	- development of an asymmetric guided bone regeneration (GBR) membrane benefiting from curcumin and aspirin	- nanofibrous asymmetric collagen/curcumin membrane	- fabrication using electrospinning technique and characterization of asymmetric membrane	- tissue and bone engineering	- The prepared membrane acts as osteoinductive material to promote the new bone formation.
Pang Z. et al., 2021 [24]	In vitro and in vivo on an animal model	- development of a bioactive coating on PEEK and investigate the effects of coating on cellular response	- Nanostructured coating of non-crystalline tantalum pentoxide (TP) on polyetheretherketone (PEEK) (PKTP).	- preparation and characterization of TP coating on PEEK by utilizing vacuum evaporation	- tissue regeneration	- Both RBMS cells and HGE cells responded strongly when exposed to PKTP with TP coating, improving surface performances.
Lam L.R.W. et al., 2021 [25]	In vitro	- development of a multifunctional core-shell nanofiber membrane	- core-shell nanofibers with encapsulated enamel matrix	- coaxial electrospinning and characterization of core-shell nanofibers.	- tissue regeneration	- Core-shell nanofiber membranes may improve outcomes in periodontal regenerative therapy.
Ma L. et al., 2021 [26]	In vitro and in vivo on a animal model	- The influence of the dosage of berberine (BBR) (25, 50, 75, and 100 μg/mL) on scaffold morphology, cell behavior, and in vivo bone defect repair were systematically studied.	- Berberine-releasing scaffold	- preparation using electrospinning technology and characterization of scaffolds	- tissue regeneration	- A BBR/PCL/COL electrospun scaffold accelerates the bone defect repair process.
Xie D. et al., 2021 [27]	In vitro	- construction of a submicro-nano structure on polyetheretherketone (PEEK) by femtosecond laser (FSL)	- submicro-nano structures on polyetheretherketone PEEK		- tissue/bone regeneration	- The 80FPK and 160FPK with submicro-nano structures significantly improved surface performances and remarkably stimulated the adhesion and proliferation of GE cells.
Zhang Y. et al., 2021 [28]	In vitro	- the effect of gold nanoparticles (AuNPs) on osteogenic differentiation of periodontal ligament stem cell (PDLSC) sheets	- gold nanoparticles (AuNPs)	- synthesis and characterization of AuNPs	- bone regeneration	- AuNPs enhance the osteogenesis of PDLSC sheets by activating autophagy.
Su T. et al., 2022 [29]	In vitro	- a new composite multifunctional coating (PHG) to improve soft tissue sealing	- a composite multifunctional coating (PHG) prepared using gelatin and polydopamine/hydroxyapatite nanoparticles (PDA-HA)	- preparation of PDA-HA-Gelatin@Ti (PHG@Ti) and surface characterization	- bone and tissue regeneration	- The proposed PHG coating may promote soft tissue sealing and bone bonding.
Kim S.Y. et al., 2022 [30]	In vitro and in vivo on an animal model	- to reduce the possibility of surgical failure caused by microbial infection	- manuka oil in a biocompatible nanostructure surface on Ti	- pure titanium with a 0.1 mm thickness coated with 0.1%, 0.5%, 1%, and 2% manuka oil	- antibacterial and anti-inflammatory capacity	- strong inhibitory effects against several pathogenic bacteria
Shi Z. et al., 2023, [31]	In vitro and in vivo on an animal model	- fabrication of FHCS hydrogels to treat the bone wound and to bridge the newborn bone tissue	- thermo-dependent hydrogel, named as FHCS	- fabrication and characterizations of the FHCS hydrogel	- tissue and bone regeneration	- An FHCS-5 hydrogel enhanced osteogenesis most significantly in the animal model of a critical-sized calvarial bone defect.
Huang A.C. et al., 2023 [32]	Animal: rats	- the therapeutic effect of nuclear factor-kappa B (NF-κB) decoy oligodeoxynucleotide (ODNs) on the extraction sockets of Wistar/ST rats	- NF-κB decoy s ODNs loaded poly(lactic-co-glycolic acid) nanospheres (PLGA-NfDs)	- preparation of decoy ODN-loaded PLGA nanosphere	- tissue and bone engineering	- The use of this nanostructure can prevent early acute inflammation in a tooth extraction socket, with the potential to accelerate new bone formation.

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
