# Peer review of "The Applicability of Nanostructured Materials in Regenerating Soft and Bone Tissue in the Oral Cavity—A Review"

_biomimetics, 2024, doi:10.3390/biomimetics9060348_

Round 1
Reviewer 1 Report
Comments and Suggestions for Authors
This review work tries to cover the very broad field of nanotechnology and nanomaterials in dentistry. While it does manage to accomplish its goal to some extent, there are some areas that can be improved. Suggestions for improvement are listed below:
- The scope of the review is not well highlighted in the introduction. There is a need to define the broad subject of nanotechnology and nanomaterials.
- The statement in the abstract about the microelectronic components is not correct. There are many that are nanoscale.
- In the introduction, it is mentioned that this review will also cover "the new techniques for obtaining these nanostructures materials". This part was not well done in this review.
- The scope of the work mentioned that both soft and hard tissues will be review. This is not clearly done throughout this review.
- Overall, there is a disconnect between what the review said it has done and what is presented as the data. Corrections need to be made.
Comments on the Quality of English LanguageNone
Reviewer 2 Report
Comments and Suggestions for Authors
The authors present a review article about oral cavity repair and featuring topics in regenerating soft and bone tissue. A summary in this field is helpful, but there are still many mistakes in this work, for this reason I suggest major revision and to address the points outlined below:
11. Line 16 and 17 has a false claim, novel CPU chips are now at 7 nm it is difficult to get 1000 times smaller
22. “the183” space missing
33. The text should be set to the format justify for easier reading and to follow the journal guidelines.
44. Line distances in paragraphs are too large
55. Line 35 reference missing
66. Line 40 reference missing
77. Line 42 reference missing
88. The claim considerable attention should be proven with 3 references not just one
99. Line 49 reference missing
110. One sentence alone is not suitable as a paragraph, should be at least 2 sentences
111. Line 65 reference missing
112. Line 67 reference missing
113. Line 72 reference missing
114. Table 1: footer of journal overlays table
115. “This The” only beginning of sentence first word title, please revise English through the text
116. A review article should have at least 50 articles. Authors could include material studies which can be in future be used for oral cavity applications.
117. Line 152 wrong reference style / no reference
118. “resultates”
119. Line 273 reference missing,
220. Line 278 here a drug delivery system is useful.1
221. The authors omitted a novel system for oral mucosa regeneration2
222. Line 293 reference missing
223. Line 302 reference missing
224. Line 305 reference missing
225. Line 318 reference missing
226. Line 364 reference missing
References
(1) Hu, N.; Frueh, J.; Zheng, C.; Zhang, B.; He, Q. Photo-Crosslinked Natural Polyelectrolyte Multilayer Capsules for Drug Delivery. Colloid. Surf., A 2015, 482, 315–323. https://doi.org/10.1016/j.colsurfa.2015.06.014.
(2) Chernova, U. V.; Varakuta, E. Y.; Koniaeva, A. D.; Leyman, A. E.; Sagdullaeva, S. A.; Plotnikov, E.; Melnik, E. Y.; Tran, T.-H.; Rutkowski, S.; Kudryavtseva, V. L.; et al. Piezoelectric and Dielectric Electrospun Fluoropolymer Membranes for Oral Mucosa Regeneration: A Comparative Study. ACS Appl. Mater. Interfaces 2024. https://doi.org/10.1021/acsami.4c01867.
Comments on the Quality of English Language
small errors
Reviewer 3 Report
Comments and Suggestions for Authors The authors produced a well-structured literature review.The introduction and objectives are clear,
although in this chapter the English seems to need improvement.
The review methods were well defined,
the results presented in an appropriate way
and the division into subchapters of both
the results and the discussion was an appropriate decision.
For these reasons, it seems to me
that this review should be accepted for publication.
I only suggest an improvement in written English. Comments on the Quality of English Language I only suggest an improvement in written English mainly in the Introduction
Round 2
Reviewer 1 Report
Comments and Suggestions for Authors
Corrections made are sufficient
Comments on the Quality of English LanguageMinor editing needed
Reviewer 2 Report
Comments and Suggestions for Authors
accept